# Impact of Androgen Deprivation Therapy Associated to Conformal Radiotherapy in the Treatment of D’Amico Intermediate-/High-Risk Prostate Cancer in Older Patients

**DOI:** 10.3390/cancers13010075

**Published:** 2020-12-29

**Authors:** Anne-Laure Couderc, Emanuel Nicolas, Romain Boissier, Mohammed Boucekine, Cyrille Bastide, Delphine Badinand, Dominique Rossi, Benedicte Mugnier, Patrick Villani, Gilles Karsenty, Didier Cowen, Eric Lechevallier, Xavier Muracciole

**Affiliations:** 1Internal Medicine, Geriatric and Therapeutic Unit, Assistance Publique-Hôpitaux de Marseille (AP-HM), and Coordination Unit for Geriatric Oncology (UCOG), PACA West, 13009 Marseille, France; benedicte.mugnier@ap-hm.fr (B.M.); patrick.villani@ap-hm.fr (P.V.); 2Medical School, Aix-Marseille University, 13005 Marseille, France; romain.boissier@ap-hm.fr (R.B.); cyrille.bastide@ap-hm.fr (C.B.); dominique.rossi@ap-hm.fr (D.R.); gilles.karsenty@ap-hm.fr (G.K.); didier.cowen@ap-hm.fr (D.C.); eric.lechevallier@ap-hm.fr (E.L.); 3Department of Medical Oncology, University Hospital of Nimes, 30000 Nimes, France; emanuel.nicolas@chu-nimes.fr; 4IDESP Institute Desbrest of Epidemiology and Public Health, Research Unit INSERM, University of Montpellier, 34000 Montpellier, France; 5Urology Department, Conception Hospital, AP-HM, 13005 Marseille, France; 6Department of Public Health, EA 3279 Self-Perceveid Health Assessment Research Unit, Medical School, Aix-Marseille University, 13005 Marseille, France; mohammed.boucekine@univ-amu.fr; 7Urology Department, Nord Hospital, AP-HM, 13015 Marseille, France; 8Radiotherapy Department, La Timone Hospital, AP-HM, 13005 Marseille, France; delphine.badinand@ap-hm.fr (D.B.); xavier.muracciole@ap-hm.fr (X.M.)

**Keywords:** localized prostate cancer, 3D conformal external beam radiotherapy, androgen deprivation therapy, older adults

## Abstract

**Simple Summary:**

Androgen Deprivation Therapy (ADT) combined with radiotherapy is recommended for a duration of 6 months in unfavorable D’Amico intermediate-risk (IR) prostate cancer and a duration of 18 to 36 months in D’Amico high-risk (HR) prostate cancer. However, as patients ≥80 years are excluded from phase III trials combining radiotherapy and ADT, there is no consensus in this population regarding the combination of these treatments. So, we aimed to report the oncological results and morbidity radiotherapy +ADT in 101 ≥ 80 years patients with IR/HR localized prostate cancers in Marseille University Hospital. The tolerance of ADT in association to radiotherapy was acceptable in this study. There was no increase in the incidence of cardiovascular events compared to the general population, whatever the duration of ADT. The absence of significant difference in biochemical recurrence-free survival or distant metastasis-free survival between 6 and 15 months of ADT in the HR group of patients raise the question of the optimal duration of ADT in this older population. However cardio-vascular evaluation and surveillance are mandatory, especially for men over 82 years old.

**Abstract:**

Purpose/objective: The association of 3D Conformal External Beam Radiotherapy (3D-CEBRT) with adjuvant Androgen Deprivation Therapy (ADT) proved to treat patients with intermediate- and high-risk localized prostate cancer (IR and HR). However, older patients were underrepresented in literature. We aimed to report the oncological results and morbidity 3D-CEBRT +ADT in ≥80 years patients. Material and Methods: From June 1998 to July 2017, 101 patients ≥80 years were included in a tertiary center. The median age was 82 years. ADT was initiated 3 months prior 3D-CEBRT in all patients, with a total duration of 6 months for IR prostate cancer (group A; *n* = 41) and 15 months for HR prostate cancer (group B; *n* = 60). Endpoints included overall survival (OS), metastasis-free survival (DMFS), biochemical recurrence-free survival (BRFS) and toxicity. Results: Five years-OS was 95% and 86.7% in groups A and B, respectively. Cardiovascular events occurred in 22.8% of ≥80 years patients with no impact on OS. In the multivariate analysis, age <82 years, Karnofsky index and normalization of testosterone levels were significantly associated with better OS. Conclusion: Age ≥80 years should not be a limitation for the treatment of IR and HR prostate cancer patients with 3D-CEBRT and ADT, but cardiovascular monitoring and prevention are mandatory.

## 1. Introduction

Prostate cancer is the most common cancer diagnosed in men in developed countries and the third cause of death in the male population [1,2]. With the increase in life expectancy, the number of older patients treated for prostate cancer is likely to increase [3]. However, patients aged 80 years and over are often excluded from phase III studies [4,5,6] making their treatment controversial. Age is considered as a factor of poor prognosis [7,8], and only a minority of patients ≥80 years receive curative treatment, even in a population considered high risk according to the D’Amico risk classification [9,10]. Theoretically, Androgen Deprivation Therapy (ADT) combined with external beam radiotherapy is recommended for a duration of 6 months [11] in unfavourable D’Amico intermediate-risk (IR) prostate cancer and a duration of 18 to 36 months in D’Amico high-risk (HR) prostate cancer [12].

However, as ≥80 years patients are excluded from phase III trials combining radiotherapy and ADT, there is no consensus in this population regarding the combination of these treatments [13,14,15]. There is a significantly higher incidence of cardiovascular diseases in this sub-group of patients [16] and data from meta-analyses suggest that ADT would considerably increase the cardiovascular risk [17,18,19]. Furthermore, as the data is primarily from populations under 80 years, the cardiovascular risk may be underestimated in older patients. Moreover, ADT can also increase the risk of cognitive disorders, osteoporotic fractures, sarcopenia, and metabolic disorders such as diabetes or dyslipidemia, which are likely to increase frailty [20,21].

In a retrospective monocentric cohort that included ≥80 years patients with intermediate and high-risk prostate cancer treated in a tertiary center, we conducted the impact of ADT in association to 3D Conformal External Beam Radiotherapy (3D-CEBRT) on the oncological outcomes and morbidity.

## 2. Results

From July 1998 to June 2017, 101 patients met inclusion criteria and underwent 3D-CEBRT + ADT for IR/HR localized prostate adenocarcinoma. The demographic, geriatric, and oncological characteristics are presented in Table 1.

The median age at diagnosis was 82 years (80–94 years). There were 41 patients with IR cancer (group A) and 60 patients with HR cancer (group B). Median PSA was 16.3 ng/mL (3.3–120 ng/mL) and 84.2% of patients had Gleason ≥7.

The majority of the population had a Karnofsky score of 100% before starting the treatment (86.2%). The most common comorbidities were cardiovascular: 43.6% had hypertension, 19.8% had coronaropathy and 13.8% had diabetes. Concerning personal drugs, 21.7% of the population had ≥ five drugs; 47.5% had anticoagulant and or/antiaggregant; 36.6% had antihypertensive drug.

Median total prostatic dose was 75.6 Gy in 42 fractions. Sixteen patients (15.8%) underwent hypofractionated radiotherapy (70 Gy in 30 fractions). The median total duration of ADT was 6 months for IR patients (group A) and 15 months for HR patients (group B).

The median follow-up was 62.8 months in the IR population (group A) and 40.5 months in the HR population (group B).

### 2.1. Overall Survival (OS)

At the date of last known follow-up, seven (17%) and ten patients (16.7%) died in groups A and B, respectively. Twelve out of the 17 deaths (70.6%) were attributed to intercurrent causes (mainly cardiovascular), two deaths were attributed to metastasis and three to unknown causes.

The overall five-year survival rate was 95% and 86.7% in groups A and B, respectively. According to the log-rank test, there was no difference in OS between groups A and B *(p* = 0.463) (Figure 1).

Multivariate analysis demonstrated that the median age <82 years (aHR = 0.11; 95% CI: 0.02–0.48, *p* = 0.003), the Karnofsky index = 100% (aHR = 0.21; 95% CI: 0.04–0.99, *p* = 0.049), and the normalization of testosterone levels (aHR = 0.13; 95% CI: 0.03–0.62; *p* = 0.010) (Table 2) were significantly associated with OS.

### 2.2. Biochemical Recurrence-Free Survival (BRFS) and Distant Metastasis-Free Survival (DMFS)

There was no PSA failure or distant metastasis in group A. PSA failure occurred in twelve patients (20%) in group B, six (10%) of which had distant metastasis. In group B, the median BRFS and DMFS were 84 months and 139 months, respectively. The BRFS and DMFS at five years were 85% and 91%, respectively. The variables associated with BRFS and DMFS are presented in Table 3. Pelvic radiotherapy was significantly associated with a shorter BRFS (OR = 5.4; 95% CI = 1.43–20.39; *p* = 0.013). HR patients who received ADT for ≤6 months (first quartile of group B) had the same BRFS (*p* = 0.632) and DMFS (*p* = 0.714) as those who received ADT for >6 months, with the same median follow-up (Figure 2a,b).

### 2.3. Adverse Events (AE)

Concerning AE related to radiotherapy, fifty-two (51.5%) and thirty-nine patients (38.6%) experienced acute urinary and gastrointestinal AE, respectively. Among them, only three patients had toxicity grade ≥3. Late genitourinary, colonic, and rectal AE were observed in forty-two (41.6%), seven (6.9%), and fifteen patients (14.9%), respectively. Only three cases of late radiation proctitis grade ≥3 (2.9%) were reported.

AE possibly related to the ADT were reported in 43.6% of the population and 13.8% were acute AE. These events are detailed in Appendix A. The most common AE were cardiovascular events that occurred in 22.8% of the population during the follow-up (median time: 35 months; ranges: 1–99 months). AE resulted in the early termination of ADT in ten patients (10%). The most common AE resulting in the termination of ADT was persistent asthenia grade ≥2, which occurred in 6.9% of patients. Concerning other toxic effects, cognitive disorders appeared in 6.9% of patients and osteoporotic fracture in 3.9% of patients during follow-up.

The median age (≥82 years) was significantly associated with the occurrence of cardiovascular events (OR = 7.15; 95% CI: 1.52–33.75; *p* = 0.013). Two patients (1.9%) died suddenly during ADT (hemorrhagic stroke and hip fracture) (Appendix A).

Following ADT, 60% and 28% of patients recovered normal testosterone levels in groups A and B, respectively (*p* = 0.003). Normalized testosterone levels were achieved after a median period of 24.9 months and 26.3 months in groups A and B, respectively. There was no significant difference in the time to normalization of testosterone levels between groups A and B (*p* = 0.714).

## 3. Discussion

This observational study reports the results of the standard treatment of patients aged ≥80 years with prostate cancer according to the international recommendations based on patients under 80 years. No PSA failure or distant metastasis was observed in ≥80 years patients with IR cancer in our cohort after a median follow-up of 62 months. In the HR group, the five-year BRFS rate and five-year DMFS rate were 85% and 91%, respectively, corresponding to the results of original radiotherapy +ADT studies [4]. Our results are consistent with a retrospective study that included 581 patients (201 patients ≥80 years) who had 3D-CEBRT + ADT for localized prostate cancer [22]. In the HR population, some patients underwent short-term ADT (first quartile of this subgroup who received ADT for 6 months or less). No difference was observed in terms of BRFS and DMFS with patients who received long-term ADT (over six months). In 2009, Bolla et al. [23] compared long-term (36 months) and short-term (6 months) ADT combined with radiotherapy to treat localized HR prostate cancer. This no inferiority trial showed inferior survival with short-term ADT with an estimated five-year mortality of 15.2% with long-term ADT compared to 19% with short-term ADT, thus establishing long-term ADT as the reference treatment for HR prostate cancer [24]. However, this trial did not include patients >80 years. Therefore, there is not currently enough data suggesting the benefits of long-term ADT in older patients.

In our study, the tolerance of older patients who receive radiotherapy and ADT was acceptable; the majority of the population was able to complete their treatment. Only a minority of the population (≤3%) experienced grade ≥3 post-radiation AE in comparison with other study [25]. Older age (>70 years) was a risk factor for increased urinary toxicity in some studies; these results contrast with our work and Goineau et al. study [26]. Acute and long-term gastrointestinal AE was acceptable; in literature, older age is a risk factor of digestive toxicity [27].

Significant cardiovascular events were observed in 22.3% of patients who underwent ADT. We did not report an increase in the incidence of major cardiovascular events contrary to previous studies [17,18,19]. The multivariate analysis demonstrated that the median age (≥82 years) was the only predictive factor of cardiovascular events.

In our cohort of patients aged ≥80 years, four reported an osteoporotic fracture (3.9%) of which one fracture during ADT that contributed to the patient’s death. An increase in the risk of osteoporotic fracture in patients treated with ADT has been described in the literature [28]. The relatively low fracture rate in our cohort could be explained by the retrospectively collection and this osteoporotic fracture rate was probably under-represented. In the literature, advanced age and the ADT duration are the two main factors associated with a fracture risk in patients receiving ADT [29]. Special attention should be paid to the bone stock in the population aged ≥80 years before initiating long-term ADT.

Androgen deprivation also has an impact on muscle and lipid metabolism with a 10% increase in body fat and a 3% decrease in lean body mass [30]. ADT can result in sarcopenia and induce asthenia and multiple difficulties doing certain exercises and movements, especially during the first year of treatment [31]. In our cohort of patients aged ≥80 years, early discontinuation of ADT was observed in 10% of the population, mainly due to debilitating, persistent asthenia grade ≥2. Moreover, radiotherapy is source of asthenia [32] (transportation to the radiotherapy center, mobility impairment, insomnia) and older men treated with both ADT and radiotherapy had higher risk to develop asthenia. The decrease in autonomy in relation to asthenia, sarcopenia, and symptoms of depression, can significantly change the quality of life of these patients [33].

An age-related physiological decrease in testosterone levels has been described in the literature and is estimated to be approximately 1% per year [34]. Among ≥80 years population, IR prostate cancer patients receiving ADT for a median duration of 6 months recovered more frequently (60% versus 28%) normal testosterone levels than those receiving ADT for a median duration of 15 months. This data suggests that age and the duration of ADT independently influence the recovery of testosterone levels. Therefore, prescribing long-term ADT in older subjects increases the risk of permanent andropause. Due to this age-related risk of permanent andropause, a decrease in the duration of ADT in older subjects with HR prostate cancer could be discussed.

In our cohort of patients aged ≥80 years, only two patients died as a result of the metastatic spread of prostate cancer. Overall survival was linked above all to non-cancer-related deaths, in majority cardiovascular causes. According to the Cox multivariate analysis, independent prognostic factors associated with OS were median age (<82 years), the Karnofsky index (= 100%), and recovery of testosterone levels. In the literature, it has already been established that the overall condition as well as comorbidities were independent factors associated with OS in cancer patients [35]. Comorbidities had no impact in our study because of lower number of comorbidities (median of Charlson score adapted to age was 5). Our cohort of patients had no specific selection, but the inclusions were realized in a long time period, which can be a bias as the management. In fact, the majority of patients were included in last ten years. Furthermore, the recovery of normal testosterone levels also seems to be significantly correlated with OS. A meta-analysis published in 2011 also reported that low testosterone levels were independently associated with an increase in all-cause mortality with an HR of 1.35 (95% CI: 1.13–1.62) [36]. This prognostic factor should be taken into account by the prescriber before initiating ADT, even though the study design did not allow directly to weight the putative detrimental role of added ADT in cardiovascular death especially in HR patients. Moreover, there does not appear to be a significant difference in BRFS and DMFS whatever the duration of ADT.

As far as we know, this is the first time the results of combined radiotherapy and ADT in a population of 101 patients aged ≥80 years and over have been reported in France. Our study has some limitations. It is a monocentric, retrospective, observational study and some adverse events that occurred during the follow-up may not have been reported in the medical records resulting in a possible underestimation of these events. Therefore it is difficult to draw conclusions from a relatively small retrospective cohort of high risk prostate cancer. Moreover, the initial testosterone levels were not determined before initiating treatment and it is, therefore, impossible to know how many patients had low testosterone levels to start with. Additionally, we do not have a control arm with patients aged ≥80 years receiving radiotherapy alone for prostate cancer. A comparative analysis carried out between patients aged ≥80 and <80 years in the same period did not highlight a significant difference in BRFS and DMFS. Another limitation of our study concerns the unavailability of some geriatric data relating to autonomy, nutritional status, and cognitive status. Access to a geriatric oncology assessment was available from 2015 for our cohort and therefore, only 14 patients out of 101 were able to undergo this geriatric oncology assessment before starting treatment. To determine the optimal treatment for patients with localized prostate cancer, the International Society of Geriatric Oncology (SIOG) recommends using the G8 screening test in subjects aged 70 and over [37,38]; patients with a G8 score ≤14/17 must be assessed by a geriatric oncologist before cancer prostate treatment.

## 4. Materials and Methods

### 4.1. Study Population and Treatment

We collected retrospectively the data from medical records of patients aged ≥80 years who underwent 3D-CEBRT for prostate cancer in the Marseille University Hospital from July 1998 to July 2017. We included patients aged ≥80 years with IR (PSA (Prostate Specific Antigen) level 10–20 ng/mL and/or Gleason score = 7 and/or clinical stage T2b) (group A) and HR (PSA >20 ng/mL and/or Gleason score ≥8 and/or clinical stage ≥T2c) (group B) prostate adenocarcinoma according to the D’Amico classification [10]. All patients had 3D-EBRT + ADT and 3D-EBRT modalities were reported in prior study [39]. All patients underwent simulation in prone position and a planning computed tomography scan was performed using 3-mm slices from iliac crest to trochanter. Physician contoured on each image prostate, seminal vesicles, bladder, and rectum with Advantage system. The 3-dimensional (3D) expansion was 10 mm around the target volumes in all directions taking into account the microscopic invasion, organ motion, and patient setup. In all patients 3-D planning was carried out with the ISIS 3-D then Pinnacle treatment planning systems. Total dose prescription followed the International Commission on Radiation Units and Measurements (ICRU) 50 guidelines and dose per fraction was 1.8 Gy in the majority of patients. The prostate and seminal vesicles were treated to 50.4 Gy followed by a boost to the prostate gland and the base of seminal vesicle (median total dose of 75.6 Gy). Androgen Deprivation therapy with injectable gonadotropin releasing hormone agonist every three months and Bicalutamide co-administered in the first 1–3 months to prevent testosterone flare was initiated 2 months before 3D-EBRT and lasted for 6 months in IR patients and 15 months in HR patients. The exclusion criteria were metastatic disease, history of prostatectomy and presence of histological variants other than adenocarcinoma.

The study was approved by a local ethics committee. All the patients were registered at baseline in accordance with the French database and Privacy law (Commission Nationale de l’Informatique et Liberté CNIL registration number: 2019-39).

### 4.2. Data Collection and Follow-Up

Overall health, comorbidities, and regular treatments were collected retrospectively from the medical records. Overall health was assessed according to the Karnofsky index [40]. We used the Charlson comorbidity index to assess comorbidities [41] and the Adult Comorbidity Assessment (ACE-27) to determine cardiovascular risk [22].

The total radiotherapy dose in Gray (Gy), the duration of ADT as well as PSA and testosterone serum levels were collected. PSA failure was defined as an increase of PSA value over PSA nadir plus 2 ng/mL after 3D-CEBRT. Distant metastasis was defined as the appearance of metastases during a relapse (whatever the type of metastasis). The long-term toxicity of radiotherapy was determined according to the Common Terminology Criteria for Adverse Events (CTCAE) v5 (2017) and reported for the entire older population (until death or last known follow-up).

To estimate the toxicity of ADT, during the follow-up (every 6-months during first 5 years and every year after) we reported the following: asthenia, locomotor events (osteoporotic fracture/arthralgia/myositis), neurological events (acute state of confusion/cognitive disorders), biological events (anemia/hepatic cytolysis), and cardiovascular events (ischemic heart disease/cardiac insufficiency/stroke/phlebitis and pulmonary embolism/carotid stenosis/sudden death or cardiovascular-related death).

#### Definition of Endpoints

Overall survival (OS) was defined as the time between the date of radiotherapy (first fraction) and the date of death or last known follow-up. Biochemical recurrence was defined by serum PSA > PSA nadir + 2 ng/mL (Phoenix Consensus Conference definition [42]). Metastatic recurrence was defined by biochemical recurrence + positive metastatic work-up.

### 4.3. Statistical Analyses

A descriptive analysis was performed to detail the main demographic, oncological, geriatric, and treatment characteristics of our population using headcounts and percentages for discrete data, as well as mean values plus or minus the standard error and the interval between the minimum and maximum values for continuous data. Cox regression analysis and the log-rank test were performed to analyze OS, DMFS, and BRFS. Variables with a *p*-value ≤ 0.1 in the univariate analysis were added to a stepwise model. A *p*-value ≤ 0.05 was considered statistically significant. All the statistical analyses were performed using SPSS software (version 17.0) for windows.

## 5. Conclusions

The tolerance of ADT in association to 3D-CEBRT is acceptable in patients aged ≥80 years. In our study, there was no increase in the incidence of cardiovascular events compared to literature, whatever the duration of ADT. The only factor associated with the occurrence of cardiovascular events was an age of over 82 years. In our cohort, the median age (<82 years) and the general condition assessed according to the Karnofsky index were the only pre-therapeutic factors independently associated with overall survival. Furthermore, after ADT, the recovery of normal testosterone levels also was correlated with overall survival. Therefore, the recovery of normal testosterone levels and the absence of significant difference in biochemical recurrence-free survival or distant metastasis-free survival between 6 and 15 months of ADT in the HR group of patients raise the question of the optimal duration of ADT. To date, there is no strong evidence in the literature to justify the use of long-term ADT in patients aged ≥80 years, even with a high risk of metastatic spread. Present multimodal strategies to manage intermediate/high risk patients are also applicable to older patients, provided that they are fit enough to receive ADT in association to 3D-CEBRT and that this condition requires patients to be submitted in advance to a geriatric and cardio-oncologic assessment.

## Figures and Tables

**Figure 1 cancers-13-00075-f001:**
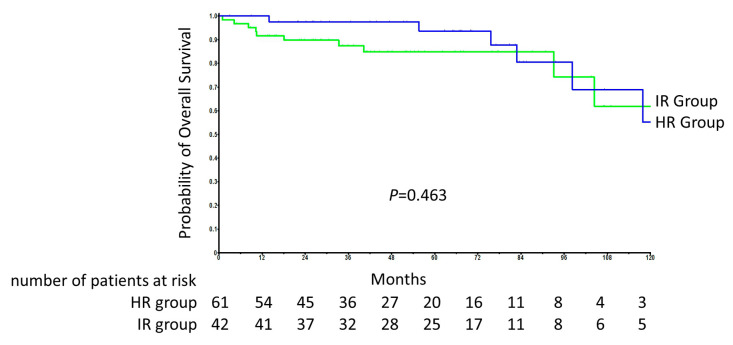
Overall survival (OS) and numbers at risk in groups A and B.

**Figure 2 cancers-13-00075-f002:**
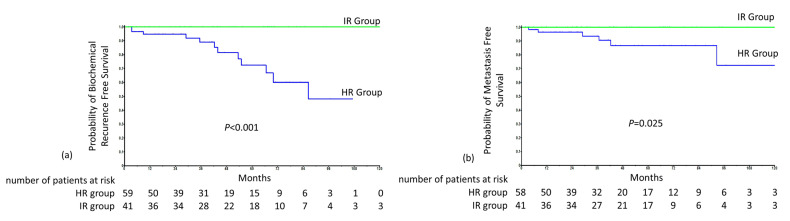
Survival in groups A and B, panel (**a**) represents biochemical recurrence-free survival (BRFS) and numbers at risk for both A and B groups, panel (**b**) represents distant metastasis-free survival (DMFS) and numbers at risk for both for both A and B groups.

**Table 1 cancers-13-00075-t001:** Demographic, medical, and oncologic characteristics of 101 older patients.

Characteristics	*n* or Median	(%) or IQR	Characteristics	*n* or Median	(%) or IQR
Age	82	80–94	Dose		
80 to 85	92	(91.0)	Median–Gy	75.6	30–80
>85	9	(9.0)	<74 Gy	9	(9.0)
Karnofsky-%-70	1	(0.9)	74–77 Gy	79	(78.2)
80	10	(10.0)	≥78 Gy	13	(12.8)
90	2	(1.9)	Androgen Deprivation Therapy (ADT) (months)		
100	87	(86.2)	No	2	(1.9)
Missing	1	(0.9)	3–6	42	(41.6)
Stage tumor			8–18	47	(46.5)
1c-2a	46	(45.6)	>18	10	(10.0)
2b	23	(22.8)	Duration of ADT-months	12	3–24
2c-4	30	(29.7)	RI	6	3–24
Tx	2	(1.9)	HR	15	4–24
Stage nodes			Charlson score	5	4–9
N0	96	(95.1)	4 to 5	62	(61.4)
N1	2	(1.9)	6 to 7	24	(23.8)
Nx	3	(3.0)	8 to 9	6	(5.9)
PSA	16.3	3.3–120	Missing	9	(8.9)
<10 ng/mL	19	(18.8)	ACE 27 score	1	0–3
10–19.9 ng/mL	49	(48.5)	0 to 1	*77*	(76.2)
>20 ng/mL	33	(32.7)	2 to 3	15	(14.8)
Gleason-<7	16	(15.8)	Missing	9	(9.0)
7 (3 + 4)	26	(25.7)	Comorbidities		
7 (4 + 3)	28	(27.7)	Hypertension	44	(43.6)
8–10	31	(30.6)	Coronaropathy	20	(19.8)
Risk according to the D’Amico classification			Diabetes	14	(13.8)
Intermediate-risk (IR)	41	(40.6)	Other cancer	15	(14.9)
High-risk (HR)	60	(59.4)	COPD	8	(7.9)
Hypofractionated radiotherapy	16	(15.8)	Polypharmacy (≥5 drugs)	22	(21.7)
Pelvic radiotherapy	16	(15.8)	Anticoagulant and/or antiaggregant	48	(47.5)
Median–Gy	46	44–50.4	Antihypertensive drug	37	(36.6)

ADT: Androgen deprivation therapy; ACE 27: Adult comorbidity assessment; COPD: Chronic obstructive pulmonary disease; HR: High-risk; IR: Intermediate-risk; PSA: Prostate specific antigen; Gy: Gray; *n*: Number.

**Table 2 cancers-13-00075-t002:** Overall survival (OS) in older patients (groups A + B; *n* = 101). Univariate and multivariate analysis.

Variables	Univariate Analysis			Multivariate Analysis		
	HR	CI 95%	*p*-Value	aHR	CI 95%	*p*-Value
Age (<82 vs. ≥82 years)	0.21	0.07–0.63	0.005	0.11	0.02–0.48	0.003
IR/HR	0.68	0.25–1.83	0.455			
ADT median duration	1.24	0.45–3.36	0.670			
Karnofsky score (=100% vs. <100%)	0.26	0.67–1.00	0.05	0.21	0.04–0.99	0.049
Charslon score (≤5 vs. >5)	1.91	1.37–11.36	0.011	1.14	0.27–4.77	0.859
ACE 27 score (≤1 vs. >1)	3.16	1.06–9.42	0.038	2.62	0.60–11.48	0.658
Anticoagulant and/or antiaggregant	1.35	0.48–3.78	0.557			
Anti-diabetes drugs	1.31	0.27–6.21	0.728			
Antihypertensive drug	3.26	1.04–10.21	0.042	1.74	0.46–6.48	0.136
Normalization of testosterone levels	0.29	0.09–0.90	0.033	0.13	0.03–0.62	0.010

ADT: Androgen deprivation therapy; ACE 27: Adult comorbidity assessment; HR: High-risk, IR: Intermediate-risk; *n*: Number; aHR: Adjusted hazard ratio; CI: Confidence interval.

**Table 3 cancers-13-00075-t003:** Biochemical recurrence-free survival (BRFS) and Distant metastasis-free survival (DMFS) in high-risk ≥80 years patients (group B; *n* = 60)**.** Univariate analysis.

Variables	BRFS			DMFS		
	OR	CI 95%	*p*-Value	OR	CI 95%	*p*-Value
Age (<82 vs. ≥82 years)	1.02	0.30–3.46	0.973	1.13	0.82–1.55	0.427
Tumoral size	1.12	0.60–2.08	0.567	2.08	0.38–11.40	0.397
Gleason score	1.41	0.16–11.95	0.747	1.18	0.35–3.93	0.779
PSA	2.14	0.62–7.41	0.228	8.99	0.95–84.48	0.055
Percentage of positive biopsy	0.86	0.22–3.25	0.826	2.44	0.25–23.75	0.442
Prostatic dose	2.07	0.4–10.79	0.384	2.11	0.24–18.16	0.496
Pelvic radiotherapy	5.4	1.43–20.39	0.013	5.54	0.98–31.08	0.052
Hypofractionated radiotherapy	4.92	0.63–37.89	0.126	11.86	0.94–148.44	0.055
ADT median duration	2.02	0.58–6.95	0.282	0.61	0.07–5.31	0.662

ADT: Androgen deprivation therapy; PSA: Prostate specific antigen; *n*: number; OR: Odd ratio; CI: Confidence interval.

## Data Availability

The data are available from the corresponding author upon reason able request.

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
