# Peer review of "Impact of Androgen Deprivation Therapy Associated to Conformal Radiotherapy in the Treatment of D’Amico Intermediate-/High-Risk Prostate Cancer in Older Patients"

_cancers, 2020, doi:10.3390/cancers13010075_

Round 1

Reviewer 1 Report

This is an intersting paper concerning the results of androgen deprivation therapy and radiotherapy in the management of intermediate/high risk prostate cancer of very elderly people (>80 years).As the authors appropriately underline,most of the trials concerning this topic which actually  provide the evidence for guide lines were not open to patients > 80 years old. Therefore contributions like the present one that focuss on elderly and very elderly patients are welcome,in spite of the limits that the authors themselves very honestly underline in the manuscript. Being that this is a retrospective analysis I will not comment on these limits ,because I understand that  data cannot be improved. However I think that data interpretation might be improved,leading to more prudent conclusions.

Major comments

1) The authors  affirm the absence of specific selection in their cohort of patients. This concept is not clear to me. Indeed patients included in the cohort were selected over a very long period of time, say 20 years ,which makes an average of 5 patients per year  and this represents a serious  selection bias. The 5-year survival data (95% and 86.7% respectively) clearly witness the fact that we are dealing with a selected "good prognosis" population, though figures are probably worse in patients older than 82, as shown by multivariate analysis. I wonder whether comparable results might have been achieved should the same approach (ie RT +ADT) be offered to the whole population of elderly patients admitted to the  Institution during the same period.

2) The low lethality of prostate cancer in this specific cohoort is also demonstrated by the fact that only 2 out of the 17 recorded deaths occurred in patients developing metastases. The other 15 deaths are likely to be attributed to CV causes. Study design doesn't allow to weight the putative detrimental role of added ADT especially in HR patients. Indeed this is possible as multivariable analysis shows that normalization of TST levels has an independent strong protective effect in this regard (HR: 0.13!!). These aspects should be addressed more properly in the discussion session.

3) It is not clear to me (but probably I missed...) on which basis the authors affirm that in their series there was no increase in the incidence of CV events compared to the general population (see conclusions)

4)I don't fully  agree with their conclusions which appear quite contradictory.In fact on one hand the authors affirm that CV events observed in their cohort was comparable to that of general population (see above...) on the other hand they suggest that in patients aged >=80 yrs, long term ADT is not justified (why since they affirm that CV events observed were comparable to that of general population...)! Now it's true that Bolla's  study  did not include patients older than 80: however it cleraly shows that 36 months are better than 6. I personally would conclude that present multimodal strategies to manage intermediate/high risk patients are applicable also to an elderly population of patients provided that  they are fit enough  to receive combined treatment, and that this condition  requires patients to be submitted  in advance to an accurate geriatric and specialistic ( namely cardio-oncologic) assessment.

Minor comments

1) The authors adopted the Phoenix criteria to define biochemical failure (i.e. PSA Progression) I wonder wether  these criteria are equally valid for patients who in addition to definitive radiotherapy receive an androgen suppressive treatment, namely for patient who received this treatment for 15 months, being that the majority of these patients did not recover a normal gonadal function( in other words remained castrated) in spite of the suspension ofADT  

Author Response

Thank you very much for the time you granted the review of our work.

Reviewer 1:

This is an interesting paper concerning the results of androgen deprivation therapy and radiotherapy in the management of intermediate/high risk prostate cancer of very elderly people (>80 years). As the authors appropriately underline, most of the trials concerning this topic which actually provide the evidence for guide lines were not open to patients > 80 years old. Therefore contributions like the present one that focus on elderly and very elderly patients are welcome, in spite of the limits that the authors themselves very honestly underline in the manuscript. Being that this is a retrospective analysis I will not comment on these limits, because I understand that data cannot be improved. However I think that data interpretation might be improved, leading to more prudent conclusions.

Major comments

  1. The authors affirm the absence of specific selection in their cohort of patients. This concept is not clear to me. Indeed patients included in the cohort were selected over a very long period of time, say 20 years, which makes an average of 5 patients per year and this represents a serious selection bias. The 5-year survival data (95% and 86.7% respectively) clearly witness the fact that we are dealing with a selected "good prognosis" population, though figures are probably worse in patients older than 82, as shown by multivariate analysis. I wonder whether comparable results might have been achieved should the same approach (ie RT +ADT) be offered to the whole population of elderly patients admitted to the institution during the same period.

Regarding your concern on, we added in discussion: Comorbidities had no impact in our study because of lower number of comorbidities (median of Charlson score adapted to age was 5). Our cohort of patients had no specific selection but the inclusions were realized in a long time period, which can be a bias as the management. But in fact, the majority of patients were included in last ten years. We did not realize any specific selection for our cohort of patients treated, but the very long time period of enrolment may represent a selection bias. However, the majority of patients were included in the last ten years

  1. The low lethality of prostate cancer in this specific cohort is also demonstrated by the fact that only 2 out of the 17 recorded deaths occurred in patients developing metastases. The other 15 deaths are likely to be attributed to CV causes. Study design doesn't allow to weight the putative detrimental role of added ADT especially in HR patients. Indeed this is possible as multivariable analysis shows that normalization of TST levels has an independent strong protective effect in this regard (HR: 0.13!!). These aspects should be addressed more properly in the discussion session.This prognostic factor must should be taken into account by the prescriber before initiating ADT, even though the study design didn't allow directly to weight the putative detrimental role of added ADT in cardiovascular death especially in HR patients.
  2. Regarding your concern on, we added in discussion: Overall survival was linked above all to non-cancer-related deaths in majority cardiovascular causes.
  3. It is not clear to me (but probably I missed...) on which basis the authors affirm that in their series there was no increase in the incidence of CV events compared to the general population (see conclusions)

Thank you for pointing this out, we clarified the sentence: compared to the general population literature

  1. I don't fully agree with their conclusions which appear quite contradictory. In fact on one hand the authors affirm that CV events observed in their cohort was comparable to that of general population (see above...) on the other hand they suggest that in patients aged >=80 yrs, long term ADT is not justified (why since they affirm that CV events observed were comparable to that of general population...)! Now it's true that Bolla's study did not include patients older than 80: however it clearly shows that 36 months are better than 6. I personally would conclude that present multimodal strategies to manage intermediate/high risk patients are applicable also to an elderly population of patients provided that  they are fit enough  to receive combined treatment, and that this condition  requires patients to be submitted  in advance to an accurate geriatric and specialistic ( namely cardio-oncologic) assessment.

To address your concern and clarify our position on ADT treatment for older adults we added in conclusion: Present multimodal strategies to manage intermediate/high risk patients are also applicable to older patients, provided that they are fit enough to receive ADT in association to 3D-CEBRT and that this condition requires patients to be submitted in advance to a geriatric and cardio-oncologic assessment.

Minor comments

  1. The authors adopted the Phoenix criteria to define biochemical failure (i.e. PSA Progression) I wonder whether these criteria are equally valid for patients who in addition to definitive radiotherapy receive an androgen suppressive treatment, namely for patient who received this treatment for 15 months, being that the majority of these patients did not recover a normal gonadal function( in other words remained castrated) in spite of the suspension of ADT  
  2. Thank you for pointing this out. Phoenix criteria are equally valid for patients receiving RT and ADT for 15 months. Phoenix criteria are equally valid regardless of ADT duration (Liu Y.M. et al. The role and strategy of IMRT in radiotherapy of pelvic tumors. Dose escalation and critical organ sparing in prostate cancer. International Journal of Radiation Oncology Biology Physics 2007; 67,1113-1123.)

Reviewer 2 Report

Comments to the author

1) General comments

This paper described the impact of RT plus ADT in patients aged ≥80 years with intermediate and high-risk prostate cancer. The tolerance of RT plus ADT in patients aged ≥80 years was acceptable. The incidence of major cardiovascular events was not increased by RT plus ADT in patients aged ≥80 years.

2) Specific comments for revision

Major

  • In figure 1 and 2, the authors should add the number at risk.

Minor

  • On page 4 in line 155. localised prostate adenocarcionma → localized prostate adenocarcinoma
  • On page 10 in line 306. localised prostate cancer → localized prostate cancer

Author Response

RESPONSE TO THE REVIEWERS:

Thank you very much for the time you granted the review of our work.

Reviewer 2:

  1. General comments

This paper described the impact of RT plus ADT in patients aged ≥80 years with intermediate and high-risk prostate cancer. The tolerance of RT plus ADT in patients aged ≥80 years was acceptable. The incidence of major cardiovascular events was not increased by RT plus ADT in patients aged ≥80 years.

  1. Specific comments for revision

Major

  • In figure 1 and 2, the authors should add the number at risk.

To address your concern on, we added number at risk on figures.

Figure 1. Overall survival (OS) and numbers at risk in groups A and B

Figures 2. Survival in groups A and B, panel (a) represents biochemical recurrence-free survival (BRFS) and numbers at risk for both A and B groups, panel (b) represents distant metastasis-free survival (DMFS) and numbers at risk for both for both A and B groups.

Minor

  • On page 4 in line 155. localised prostate adenocarcionma → localized prostate adenocarcinoma
  • On page 10 in line 306. localised prostate cancer → localized prostate cancer

To address your concern on, we made the required corrections

Reviewer 3 Report

Couderc et al. present a retrospective analysis of outcomes of older patients with prostate cancer treated with radiotherapy plus ADT.  The outcomes for ADT+EBRT in this population is not well known due to lack of inclusion in clinical trials.  They describe oncologic outcomes.  However, one of the major missing discussion points really underlies the reason that we do not have much literature or clinical trial data in this population -- ultimately the impact of prostate cancer on survival in older men is limited when considering life expectancy and competing causes of death. In fact, current NCCN guidelines, for example, have observation as the preferred approach in men with a <10 year life expectancy and a diagnosis of unfavorable intermediate risk prostate cancer.  This is derived from the studies showing that survival is not impacted for a decade of longer after local treatment for prostate cancer, and the ability to palliate prostate cancer later when other comorbidities or limitations of life exist.  In the US, for example, the social security administration actuary life tables for men 80 years old have a ~8 year life expectancy.  Most men would not be treated with ADT + EBRT based upon competing mortality risk.  Therefore the cohort presented here most likely to be impactful to the literature is the high risk group.  It is difficult to draw conclusions from a relatively small retrospective cohort of high risk prostate cancer here, given inherent limitations of a retrospective study spanning the ~20 period presented here.

Author Response

RESPONSE TO THE REVIEWERS:

Thank you very much for the time you granted the review of our work.

Reviewer 3 :

Couderc et al. present a retrospective analysis of outcomes of older patients with prostate cancer treated with radiotherapy plus ADT. The outcomes for ADT+EBRT in this population is not well known due to lack of inclusion in clinical trials. They describe oncologic outcomes. However, one of the major missing discussion points really underlies the reason that we do not have much literature or clinical trial data in this population -- ultimately the impact of prostate cancer on survival in older men is limited when considering life expectancy and competing causes of death. In fact, current NCCN guidelines, for example, have observation as the preferred approach in men with a <10 year life expectancy and a diagnosis of unfavorable intermediate risk prostate cancer. This is derived from the studies showing that survival is not impacted for a decade of longer after local treatment for prostate cancer, and the ability to palliate prostate cancer later when other comorbidities or limitations of life exist. In the US, for example, the social security administration actuary life tables for men 80 years old have a ~8 year life expectancy. Most men would not be treated with ADT + EBRT based upon competing mortality risk.

Thank you for pointing this out. In a large cohort study including more than 120 000 men aged 55-95 years, old age at diagnostic was associated with poorer prognosis treated with deferred treatment, androgen deprivation therapy or radical prostatectomy (A.Petterson et al, Age at diagnosis and prostate cancer treatment and prognosis; a population-based cohort study. Ann Oncol 2018 29: 377-385,2018). Their findings argue against a strong inherent effect of age on risk of prostate cancer death, but indicate that in current clinical practice, old men with prostate cancer receive insufficient diagnostic workup and subsequent curative treatment.

Therefore the cohort presented here most likely to be impactful to the literature is the high risk group. It is difficult to draw conclusions from a relatively small retrospective cohort of high risk prostate cancer here, given inherent limitations of a retrospective study spanning the ~20 period presented here.

Thank you for pointing this out, we added in limitations: Therefore It is difficult to draw conclusions from a relatively small retrospective cohort of high risk prostate cancer

Round 2

Reviewer 3 Report

Authors have addressed the reviewers concerns and this study reasonably adds to literature in population with paucity of data.